# Cell Models of Castration Resistant and High Dose Testosterone-Resistant Prostate Cancer Recapitulate the Heterogeneity of Response Observed in Clinical Practice

**DOI:** 10.3390/cancers17040593

**Published:** 2025-02-10

**Authors:** Laura S. Graham, Lih-Jen Su, Andrew Nicklawsky, Frances Xiuyan Feng, David Orlicky, Joseph Petraccione, Maren Salzmann-Sullivan, Steven K. Nordeen, Thomas W. Flaig

**Affiliations:** 1Division of Medical Oncology, Department of Internal Medicine, University of Colorado Denver Anschutz Medical Campus, Aurora, CO 80045, USA; lih-jen.su@cuanschutz.edu (L.-J.S.); joseph.petraccione@cuanschutz.edu (J.P.); maren.salzmann-sullivan@cuanschutz.edu (M.S.-S.); thomas.flaig@cuanschutz.edu (T.W.F.); 2Biostatistics and Bioinformatics, University of Colorado Cancer Center, Aurora, CO 80045, USA; andrew.nicklawsky@cuanschutz.edu; 3Department of Pathology, University of Colorado Denver Anschutz Medical Campus, Aurora, CO 80045, USA; frances.feng@cuanschutz.edu (F.X.F.); david.orlicky@cuanschutz.edu (D.O.); steve.nordeen@cuanschutz.edu (S.K.N.)

**Keywords:** prostate cancer, supraphysiologic testosterone, cell models, testosterone resistance, androgen deprivation therapy

## Abstract

High dose testosterone therapy is emerging as a viable treatment strategy for metastatic castration-resistant prostate cancer. Although a significant minority of patients have a profound response to testosterone therapy, a majority show no response or have subsequent progression, highlighting the need for novel therapeutic strategies to enhance the efficacy of high dose testosterone therapy. We have developed heterogeneous cell models of castration-resistant and high dose testosterone-resistant prostate cancer that will allow for the identification of mechanisms of resistance and provide a pre-clinical vehicle for pharmacologic testing for this emerging clinical entity.

## 1. Introduction

Despite major therapeutic advances over the last decade in the treatment of metastatic prostate cancer, it remains the second leading cause of cancer death among American men [1,2]. In its initial stages, prostate cancer is highly reliant on androgen receptor signaling for cell growth and proliferation, and the mainstay of treatment of advanced disease is androgen deprivation therapy (ADT) [3]. Resistance to ADT ultimately develops, resulting in a lethal phenotype known as castration-resistant prostate cancer (CRPC). Despite recent drug approvals, treatment options for CRPC remain limited [4,5,6,7,8,9,10,11].

Perhaps counterintuitively, some men with CRPC respond to supraphysiologic doses of testosterone. One way this is employed is via bipolar androgen therapy (BAT), in which serum testosterone levels cycle between supraphysiologic ranges and near-castrate levels with monthly testosterone injections [12]. In the Phase II TRANSFORMER trial, 28.2% of patients with CRPC who had previously progressed on the second-generation AR-targeted agent abiraterone acetate experienced a decline in PSA of 50% or more from pre-treatment baseline with BAT. Strikingly, BAT appeared to re-sensitize men to AR-targeted therapy, with 77.8% of men experiencing a PSA50 response to the second-generation AR antagonist enzalutamide after BAT [13]. Similar results were reported in a small cohort of men who had previously progressed on enzalutamide [14]. Even with these successes, however, progression with supraphysiologic testosterone appears to be inevitable over time, highlighting the need for deepening the understanding of therapeutic resistance in both high dose testosterone-naïve CRPC and high dose testosterone-resistant CRPC. An increased understanding of these drivers of testosterone resistance will open the door to novel treatment strategies for clinical assessment, either in combination with supraphysiologic testosterone therapy or after progression. In addition, more studies are needed to identify markers of high dose testosterone susceptibility.

A significant challenge in the development of novel therapeutics in the late stages of CRPC is tumor heterogeneity both within and between patients. Successful models of in vitro and in vivo CRPC therefore need to replicate this diversity. We have developed three cell models of high dose testosterone-sensitive CRPC and high dose testosterone-resistant CRPC which recapitulate the heterogeneity seen in patients with CRPC. We demonstrate the response to both testosterone treatment and testosterone withdrawal of these cell lines in vitro and in vivo and illustrate that response is correlated with overall levels of AR expression and the heterogeneity of expression within a tumor population. This work lays the foundation for future experiments testing new treatments and treatment combinations for high dose testosterone-resistant CRPC.

## 2. Materials and Methods

### 2.1. Cell Lines

All cell lines were derived from the prostate cancer cell line LNCaP which was obtained from the University of Colorado Cancer Center (UCCC, Aurora, CO, USA) Tissue Culture Shared Resource and were authenticated via single tandem repeat analysis. The selection of LNCaP variants was described previously [15]. The P1 cell variant was selected via repeated passage and grown in a low androgen medium. The M1 and M2 cell lines were derived through the initial selection for growth in a medium containing the androgen antagonist enzalutamide followed by selection in a low androgen medium. From each of the 3 CRPC cell variants, an HTR resistant line was selected via repeated passage in a low androgen medium supplemented with 1 nM R1881 (synthetic androgen). These lines were designated as P1R1/R, M1R1/R, and M2R1/R.

### 2.2. Clonogenic Assay

Clonogenic assays were performed by plating cells in 24-well cell culture plates in media supplemented with charcoal-stripped serum (CSS) and the indicated drug concentrations. Cells were plated at 2000 cells per well, and all were grown for at least 14 days. When the first well became confluent, the entire plate was fixed and analyzed. Each plate had only one cell line, and each cell plate was normalized to its own control. Each condition was assessed in quadruplicate wells. Crystal violet stains were analyzed with the ImageJ software (https://imagej.net/ij), version 1.51j8, Java 1.8.0_112 (64-bit), and the colony area was analyzed with an ImageJ plugin (https://journals.plos.org/plosone/article?id=10.1371/journal.pone.0092444 (accessed on 8 March 2022)).

### 2.3. In Vivo Tumor Implantation

Athymic nude male mice (Jackson and Envigo laboratories, Bar Harbor, ME, USA) were housed at the University of Colorado Animal Center and surgically castrated at 6–8 weeks of age. Tumor cells were implanted in animals anesthetized with inhaled isoflurane/oxygen via the subcutaneous injection of 2,000,000 cells in 100 µL of Matrigel into both flanks using a 27–30 g needle. Tumor cells were implanted either one week after castration for experiments with the CRPC cell lines or one week after testosterone pellet implantation for experiments with the HTR cell lines.

### 2.4. Testosterone Pellet Implantation

Following the administration of buprenorphine ER pain medication, a 12.5 mg slow-release testosterone pellet (Innovative Research of America, Sarasota, FL, USA, NA-151) was injected subcutaneously approximately one week after surgical castration for experiments with the HTR lines. This was performed because a preliminary experiment with P1R1/R indicated that the tumors would not implant efficiently in mice with physiologic testosterone levels. Immunocompromised mice exhibit low levels of testosterone [16]. For experiments with the CRPC lines, testosterone pellets were administered after tumors were established and palpable, approximately 300–500 mm^3^. Using a size 10 trocar, an entry was made with a posterior nick, and the pellet was implanted subcutaneously between the shoulder blades. Trocar incision was closed using Vetbond tissue glue. If mice remained alive 90 days after the implantation of testosterone pellet, a new pellet was inserted after the removal of the old pellet when feasible. In select cases, it was not feasible to remove the old pellet due to the degradation of the pellet due to tissue growth around the pellet.

### 2.5. Testosterone Pellet Removal

Mice were anesthetized prior to testosterone pellet removal with isoflurane. Extended-release buprenorphine was used for pain relief. An incision was made on the back of the neck above the site of pellet implantation, and the pellet was excised. The incision was closed using a single wound clip which was removed upon healing after 7–10 days.

### 2.6. Tumor Volume Measurement

Tumor measurements were recorded using Bluetooth calipers. Measurements were taken twice per week. Tumor volume represents the summed tumor volume when tumors were present in both flanks.

### 2.7. Tissue Samples and Immunohistochemistry

In order to quantify AR expression in each cell line, histologic analyses were performed on formalin-fixed tissues. Sections with a thickness of 5 µM were used. The sources of primary and secondary antibodies and methodological details for their use are as follows: Androgen Receptor (Cell Marque, Rocklin, CA, USA, #200R-15) 1:200, Secondary: (EnvisionPlus Rabbit, Dako, Glostrup, Denmark, # K4003), Ki67 (Thermo Scientific, Waltham, MA, USA, #RM-9106-S1) 1:400, and Secondary: Horse anti-rabbit (Vector Labs, Burlingame, CA, USA, # MP-7401). Slides from 3 to 6 animals bearing tumors from each of the CRPC and HTR cell lines were sent in a blinded fashion to an expert pathologist (F.F.). Tumor areas were randomly selected, photographed, and printed using a color printer. Stromal cells were excluded from this study. AR immunoreactivity was scored using the H-score system, where the density of staining (‘0’ for none, ‘1’ for faint, ‘2’ for intermediate, and ‘3’ for prominent) was multiplied using the percentage of cells at each level, with a resultant total H-score ranging from 0 to 300. A total of 28 slides were assessed, with a mean of 5363 cells scored per slide (range 4707–7080).

### 2.8. Statistical Methods

Colony growth was normalized relative to the mean percent growth observed under baseline conditions (no synthetic androgen). Differences between androgen concentrations and baseline were assessed using the Wilcoxon rank-sum exact test. Tumor growth volume was primarily explored using descriptive statistics and was summarized by treatment status and cell line. Overall survival as defined as days from testosterone implantation or matched control was calculated by treatment group using the Kaplan–Meier method. Differences in survival between the groups were assessed through the log-rank test or through two-stage hazard rate comparison in the case of violation of the proportional hazards assumption [17]. Median survival and 95% confidence limits are reported; however due to the small sample size, the upper bounds could not be determined because of computational instability. H-scores were summarized by group (testosterone-sensitive CRPC vs. HTR CRPC), and descriptive statistics were calculated. A Wilcoxon rank-sum exact test was performed to examine the differences between the groups. To further understand the differences between the groups by accounting for clustering effects among slides within the cell line, a mixed model with random intercepts was fit. To explore the raw results, chi-square tests were performed between matched lines in different groups by examining the distribution of cells by intensity. Analyses were performed using R version 4.3.1 (R Core Team, Vienna, Austria), and an alpha level of 0.05 was used to assess statistical significance.

## 3. Results

### 3.1. In Vitro Modeling of High Dose Testosterone Resistance

Each of the three CRPC cell lines (M1, M2, and P1) was plated in a low androgen medium and then treated with R1881, a synthetic androgen, at various concentrations (Figure 1) [15]. The three CRPC cell lines exhibited a reduction in colony growth with R1881, although this was not statistically significant. The M1 cell line had slight nonsignificant stimulation at a lower R1881 concentration (0.03 nM) but had a reduction at all other concentrations tested, although this did not reach statistical significance.

An HTR cell line was selected from each one of the CRPC lines via repeated passage in a medium supplemented with 1 nM of R1881, designated as M1R1/R, M2R1/R, and P1R1/R, respectively. As shown in Figure 1, all three cell lines grew as well in high androgen (1 nM R1881) as they did in the medium without androgen supplementation. Once again, the dose response to hormone differed among the three variants. The growth of both M1R1/R and M2R1/R was stimulated by lower levels of androgen, whereas the growth of P1R1/R was unaffected at any level of androgen up to 1 nM.

### 3.2. In Vivo Modeling of High Dose Testosterone Response in CRPC Models

To test whether these cell models selected in vitro would exhibit accurate representations of both CRPC and HTR prostate cancer in vivo, we analyzed tumor volume and overall survival in mice injected with these cell lines and subjected to hormonal manipulation. Each of the three independently derived cellular models of CRPC (M1, M2, and P1) was injected into castrated male mice. Ten (five treated, five untreated), seventeen (eight treated, nine untreated), and twelve (six treated, six untreated) mice were analyzed in each cohort, respectively. Tumors were allowed to establish. When tumor volume reached 300–500 mm^3^, testosterone pellets were implanted. A tumor size-matched animal that did not receive a testosterone pellet was paired with an animal that received the hormone implant. Tumor volume was measured before and twice per week after the implantation of a testosterone pellet. Figure 2 shows the tumor volume at Day 30 from the beginning of testosterone treatment (or day of sacrifice due to tumor burden). The three models showed a wide range of responses to the testosterone treatment. P1 tumors grew rapidly in castrated mice, and growth was strongly inhibited by the testosterone treatment. In some cases, tumors shrank to a marginally palpable size. In contrast, M1 and M2 tumors grew slower and had more variability in response.

Overall survival of the mice implanted with these three CRPC cell lines is shown in Figure 3, depicted as days from testosterone implantation or matched control. Mice that received the testosterone treatment in the M2 (Median 95.5 months, 95% CI, 91-NA) and P1 (Median 104.5 months, 95% CI, 38-NA) cell lines had a statistically significant improvement in overall survival (*p* = 0.03 and *p* = 0.01, respectively). Survival was prolonged in the M1 (Median 76 months, 95% CI, 37-NA) cell line with the testosterone treatment, although this difference did not reach statistical significance (*p* = 0.53). M1 and M2 exhibited a separation of survival curves around 50–60 days, whereas the separation in survival was much more pronounced in P1, occurring between 20 and 30 days, likely reflective of both the rapid growth of P1 cells and their exquisite response to the testosterone treatment. Two of six mice exhibited a remarkably durable response with survival on the order of one year. One death in the testosterone-treated P1 tumors was due to an unknown cause (not related to tumor size necessitating euthanasia).

### 3.3. In Vivo Modeling of High Dose Testosterone Resistance

We then assessed whether the CRPC models that were selected in vitro for resistance to androgens exhibited an HTR phenotype in vivo. We injected our three models of HTR (M1R1/R, M2R1/R, and P1R1/R) into the flanks of different cohorts of castrated male mice in the presence of a testosterone pellet. Eight (four with testosterone removed and four with testosterone replaced), eleven (four and seven), and nine (three and six) mice were analyzed in each cohort, respectively. Mice were paired based on similar tumor volume, and one of each pair had the testosterone pellet removed, while the other one had the testosterone pellet remain. If there were survivors past 90 days, the testosterone pellet was replaced. Figure 4 shows tumor volume in all three HTR models with the remaining testosterone pellet and the removed testosterone pellet. There was notable variability between models, although some mice in all models had tumor shrinkage in response to testosterone pellet removal. P1R1/R appeared to have the most response to testosterone removal. Overall survival is shown in Figure 5, which illustrates modestly improved survival in all three models of HTR when the androgen environment was altered via the removal of the testosterone pellet. However, in none of these small cohorts of the three HTR models did the observed differences in overall survival achieve statistical significance.

### 3.4. Assessment of Androgen Receptor Expression

Androgen receptor signaling has been postulated to play an important oncogenic role in the development of castration resistance and subsequently to the responsiveness to supraphysiologic levels of testosterone [18,19]. We hypothesized that there would be significant differences in AR expression between both castrate-resistant and high dose testosterone-resistant cell lines, as well as between each individual cell line model. We quantified AR expression in each cell line by calculating an H-score from randomly chosen slides from animal tumors derived from each cell line. The average (standard deviation) H-score for M1, M2, and P1 was 87.8 (22.5), 130.5 (27.2), and 218.8 (41.5), respectively. The average (standard deviation) H-score for M1R1/R, M2R1/R, and P1R1/R was 85.5 (17.9), 84.0 (14.8), and 122.8 (19.5), respectively. The mean H-score for the high dose testosterone-sensitive CRPC lines was 145 (62) compared to 97 (25) for the high dose testosterone-resistant CRPC lines. When accounting for repeat measures by slide, this difference was not statistically significant; this was unsurprising given the high variation between cell lines in each group. There is a correlation between AR expression levels, consistency (H-value) in a given variant, and its ability to respond to testosterone therapy (P1 > M2 > M1 overall survival, Figure 3).

In the HTR lines there was much less variability in AR intensity compared to the CRPC lines. To further analyze the levels and heterogeneity of AR expression in tumors derived from the three CRPC and HTR variants, the cell-to-cell distribution of AR staining is shown in Figure 6. In the P1 model of CRPC, the variant most responsive to high testosterone therapy, over 85% of the cells, was scored as 2+ or 3+ for AR expression. P1 also had, by far, the highest H-score, 218.8. M2 exhibited the next highest level of 2+ plus 3+ staining at 39.3% and had an intermediate H-score of 130.5. M1, the CRPC variant least responsive to high androgen therapy in vivo, displayed the fewest (0.3%) 3+ expressors. Over 90% of the population scored negative or was faint for AR expression, an H-score of 87.8. In vitro selection of CRPC lines for androgen resistance shifts the distribution toward lesser AR expression in the P1 and M2 lineages but, interestingly, not in the M1 lineage. Whatever the hormonal environment, be it low or high androgen, there was always a population with the highest expressors and cells without AR expression that survived hormonal manipulations. The maintenance of this heterogeneity most likely contributes to the ability of the tumor cells to survive and adapt to hormonal manipulation of their environment.

## 4. Discussion

As the use of supraphysiologic testosterone to treat CRPC increases in clinical practice, further studies will be needed to identify the molecular mechanisms of resistance and combination treatment strategies. A particular challenge will be capturing the differing real-world responses to testosterone therapy in men with CRPC. Response rates to supraphysiologic testosterone therapy range from 17 to 80% in reported clinical trials and with high variability in the duration of response [13,14,20,21,22,23]. To begin to address this challenge of creating a heterogeneous model of high dose testosterone resistance, we have developed three independently derived, paired models of high dose testosterone-sensitive CRPC and high dose testosterone-resistant CRPC. We have shown through their in vitro and in vivo response to high dose testosterone treatment or withdrawal, as well as their levels of AR staining, that these cell lines mimic the heterogeneity seen in patients, which provides a useful tool for drug development and preclinical studies in this understudied and emerging disease state.

One of the most common mechanisms of resistance developed by prostate cancer cells in response to castrate conditions is an increase in AR protein expression [19,24,25,26]. If these high AR-expressing cells are exposed to supraphysiologic testosterone, ligand binding to the AR is saturated and accumulates in the nucleus at a high enough level to stifle proliferation and induce cell death [24]. This is seen most dramatically in the P1 model that exhibits the highest H-score for AR expression where testosterone therapy of an established tumor extended the life span by as much as almost one year. The M2 line expressed more AR than M1 and less than P1. It exhibited an overall survival response intermediate to that of P1 and M1. The M1 line had very few high (3+) AR expressors, but almost 80% expressed some AR. Tumor volume was suppressed effectively as seen in M2 tumors at 30 days of treatment, but that suppression was not maintained. Overall survival was only marginally improved with the testosterone treatment. These results support the contention that uniformly higher levels of AR expression correlate with the best response to high androgen therapy. Lower AR levels and increased heterogeneity of AR expression are associated with a transient or no response to such therapy.

The selection of HTR variants from the three cell line models in cell culture results in cell lines that can grow tumors robustly in testosterone-supplemented, castrated, and athymic mice. The response of the tumors derived from the HTR lines to the withdrawal of testosterone supplementation is modest. While a change in overall survival is seen with all three HTR variants, in no case with the small cohorts is increased survival statistically significant. Initial clinical results with men who have developed resistance to high androgen therapy indicate that these men often respond favorably to treatments with the androgen antagonist enzalutamide [13]. This suggests that adaptation to high androgen therapy can re-sensitize tumors to further hormonal manipulation, albeit with a likelihood of a decreased efficacy. Resistance to androgen deprivation therapy can develop in a myriad of other ways beyond the androgen receptor as well. Several subtypes of androgen indifferent prostate cancer have been described in the literature, most notably, neuroendocrine (NE) prostate cancer, although phenotypes that lack NE features but also lack or are low in AR expressed genes have also been described [27,28]. Although it has previously been described that culturing LNCaP cells in hormone-depleted media or with enzalutamide can result in neuroendocrine transdifferentiation, we did not see any histologic features of such a transformation in our cell lines [29,30].

We demonstrate that there are differences at the gene and tumor level between each cell line within both the HTR and testosterone-sensitive groups. Future experiments will aim to further characterize the genomic, transcriptomic, and metabolomic characteristics of these cell lines. This information will allow us to hypothesize and test new treatment strategies for high dose testosterone resistance, which we anticipate will become an important emerging clinical phenotype as the use of high dose testosterone in CRPC becomes more widely adopted. There are already several recently completed and ongoing clinical trials testing high dose testosterone (e.g., NCT04704505, NCT05011383, NCT04363164, NCT03734653).

## 5. Conclusions

In summary, we have developed several models of both CRPC and high dose testosterone-resistant prostate cancer, which recapitulate the clinical heterogeneity seen in patients with this disease. These models can be used to further understand the disease biology of high dose testosterone resistance and hypothesize novel treatment approaches. We have ongoing experiments designed to further characterize the molecular drivers of these HTR models. Additionally, we have ongoing drug development studies to identify treatment options for further clinical study.

## Figures and Tables

**Figure 1 cancers-17-00593-f001:**
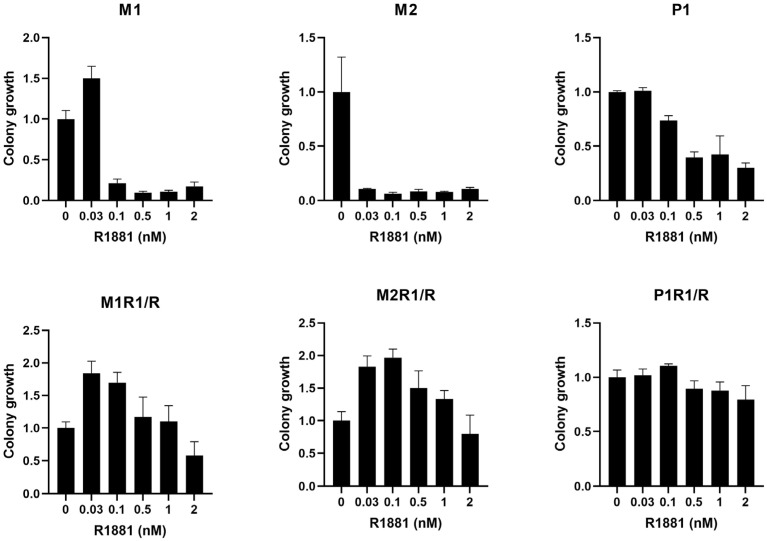
Clonogenic assay results of CRPC lines (M1, M2, and P1) and HTR lines (M1R1/R, M2R1/R, and P1R/1) incubated with synthetic androgen R1881.

**Figure 2 cancers-17-00593-f002:**
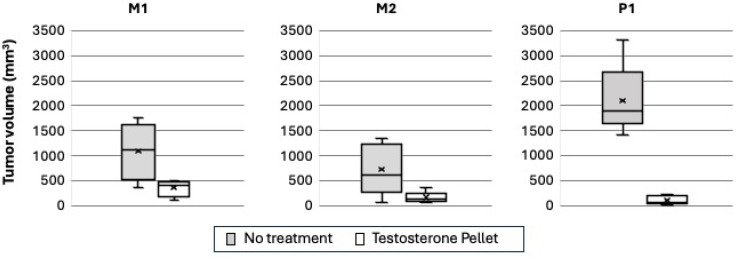
Castrate-resistant prostate cancer cell lines demonstrate varying responses to testosterone pellet implantation in vivo. Tumor volumes with and without the testosterone pellet are shown at Day 30 or at time of mouse euthanasia due to tumor growth if this occurred prior to Day 30. The black center line denotes the median value (50th percentile), and the surrounding box contains the 25th to 75th percentiles. The black whiskers mark the 5th and 95th percentiles, and the x represents the mean.

**Figure 3 cancers-17-00593-f003:**
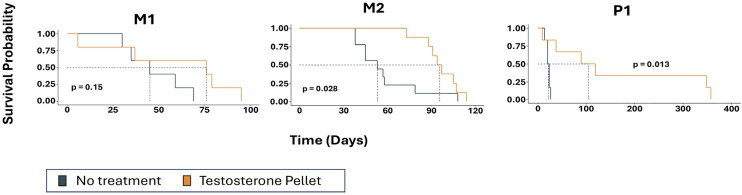
Overall survival for mice implanted with CRPC tumor cell variants, M1, M2, or P1, and then treated with the testosterone pellet (orange) or no treatment (black). Note that the *x*-axis varies for each variant. Day zero indicates the beginning of the treatment with testosterone or pairing in the case of untreated controls.

**Figure 4 cancers-17-00593-f004:**
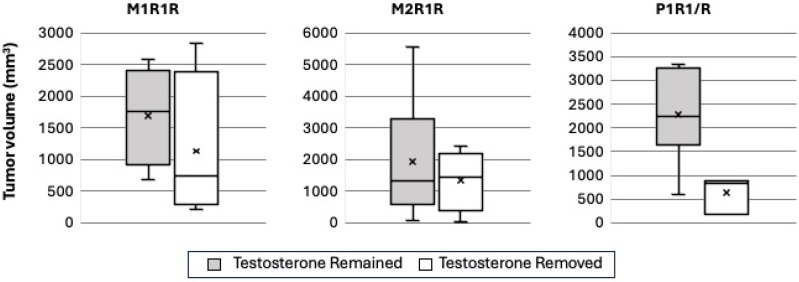
High dose testosterone-resistant prostate cancer cell lines demonstrate varying responses to testosterone pellet removal in vivo. Tumor volumes with and without the testosterone pellet are shown at Day 30 or at time of mouse euthanasia due to tumor growth if this occurred prior to day 30. The black center line denotes the median value (50th percentile), and the surrounding box contains the 25th to 75th percentiles. The black whiskers mark the 5th and 95th percentiles, and the x represents the mean.

**Figure 5 cancers-17-00593-f005:**
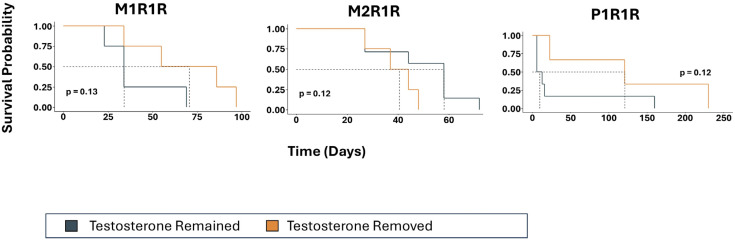
Overall survival for mice implanted with high dose testosterone tumor cell variants, M1R1R, M2R1R, and P1R1R, in the presence of a testosterone pellet. Mice were then treated with testosterone removal (orange) or the maintenance of testosterone (black). Day zero indicates the start of testosterone removal or pairing in the case of controls with testosterone pellet remaining.

**Figure 6 cancers-17-00593-f006:**
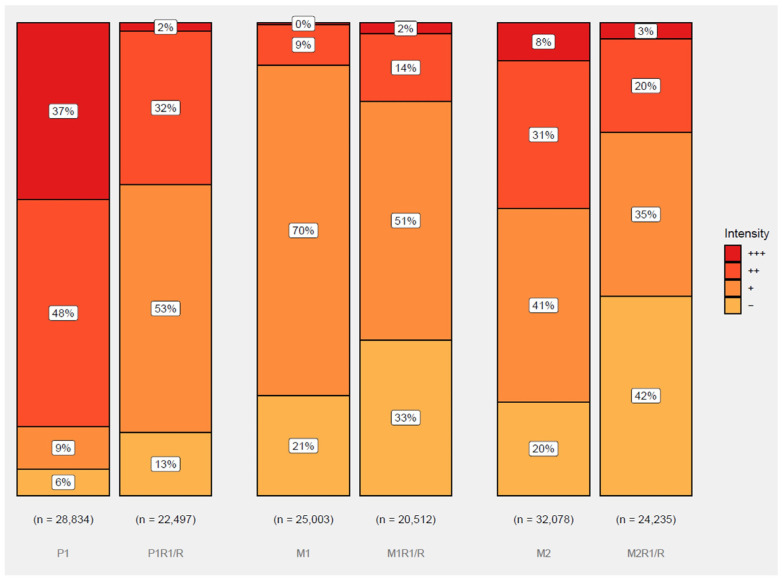
Levels of androgen receptor (AR) in parental CRPC models and their high dose testosterone derivative. Cells were scored based on the intensity of AR staining (0–3+). The intensity is denoted in the figure as – for 0, + for 1+, ++ for 2+, and +++ for 3+. The left hand column of each pair shows the distribution of intensity staining for parental cell lines (P1, M1, and M2), and the right hand columns show the distribution of intensity staining for high dose testosterone-resistant cell lines (P1R1/R, M1R1/R, and M2R1/R).

## Data Availability

All data are available upon request.

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
