# Peer review of "Cell Models of Castration Resistant and High Dose Testosterone-Resistant Prostate Cancer Recapitulate the Heterogeneity of Response Observed in Clinical Practice"

_cancers, 2025, doi:10.3390/cancers17040593_

Round 1

Reviewer 1 Report

Comments and Suggestions for Authors

In this original research article by Graham et al, the development of LNCaP-derived CRPC cell lines that are either sensitive (M1, M2 and P1) or resistant (M1R1/R, M2R1/R, P1R1/R) to high dose testosterone are reported. The authors highlight response metrics of the cell lines to R1881 in vitro (clonogenic assays) and high dose testosterone in vivo (tumor volume and survival). Additionally, immunohistochemistry of AR expression from CRPC and HTR xenografts suggests that level of AR expression underpins the response to therapy. While the findings and generation of new preclinical models are potentially interesting, limited molecular characterization of the reported CRPC and HTR cell lines significantly detracts from the translatability of the models to the patient population and the synthesis of author conclusions. Below are comments and suggestions:

-Figure 1 is challenging to interpret given the information provided in the figure legend and Materials and Methods. Please clarify the cell lines in the clonogenic assay that were “supplemented with fetal bovine serum (FBS) or charcoal-stripped serum (CSS)" (Page 2, Line 81).  Were the assay parameters between the cell lines consistent? Additionally, please indicate the duration of R1881 treatments.

-Please provide statistical analyses between treatment groups in Figure 1 to support your claims for all lines tested. The authors state that “The three CRPC cell lines all exhibited a dose-dependent reduction in colony growth with R1881” (page 4, Line 148). However, the M1 cell line appears to be stimulated by R1881 at 0.03 nM as reported in the M1R1/R cell line. The M2 cell line is exquisitely sensitive to R1881 with no apparent response differences between all R1881 concentrations tested.

-Please clarify the number of animals in each experimental group in figure legends or graphs for Figures 2-5. The reader must infer the number of animals in each of the groups from total number of animals reported in the main text.

 -Figure 6, please describe the analytical tools or approaches used to determine cell numbers and estimate percentage of AR+ cells from IHC images. Are stromal cells excluded from the analyses or are they contributing to the data interpretations?

-Culturing LNCaP cells in hormone depleted media, such as the P1 line, has been reported to induce cellular plasticity and neuroendocrine features (eg. PMID: 21227062; 27403603). LNCaP cells cultured in enzalutamide, such as M1 and M2, have been reported to express AR splice variants (eg LNCaP95 cells). Additional molecular phenotypes of AR+ and AR- CRPC have also been reported in the patient population (i.e. amphicrine, double-negative, AR-low). Moreover, the downregulation of AR protein expression observed in cells from the three HTR cell lines potentially indicates further transdifferentiation to other molecular subtypes or expression of AR-splice variants not detected by the IHC antibody. Do western blots, qPCR or IHC of neuroendocrine markers, EMT markers or AR-V7 and target genes provide more insights into molecular characteristics of CRPC lines reported here (P1, M1 and M2) or the mechanisms of resistance to high dose testosterone in the HTR lines (P1R1/R, M1R1/R and M2R1/R)?

Author Response

We thank the reviewers for taking the time to review our manuscript. We have responded to each review below in addition to including a manuscript with track changes in the resubmitted files.

Point-by-point response to Comments and Suggestions for Authors (Reviewer 1)In this original research article by Graham et al, the development of LNCaP-derived CRPC cell lines that are either sensitive (M1, M2 and P1) or resistant (M1R1/R, M2R1/R, P1R1/R) to high dose testosterone are reported. The authors highlight response metrics of the cell lines to R1881 in vitro (clonogenic assays) and high dose testosterone in vivo (tumor volume and survival). Additionally, immunohistochemistry of AR expression from CRPC and HTR xenografts suggests that level of AR expression underpins the response to therapy. While the findings and generation of new preclinical models are potentially interesting, limited molecular characterization of the reported CRPC and HTR cell lines significantly detracts from the translatability of the models to the patient population and the synthesis of author conclusions. Below are comments and suggestions:

Comment1 :Figure 1 is challenging to interpret given the information provided in the figure legend and Materials and Methods. Please clarify the cell lines in the clonogenic assay that were “supplemented with fetal bovine serum (FBS) or charcoal-stripped serum (CSS)" (Page 2, Line 81).  Were the assay parameters between the cell lines consistent? Additionally, please indicate the duration of R1881 treatments.

We thank the reviewers for taking the time to review our manuscript. We have responded to each review below in addition to including a manuscript with tracked changes.

Reviewer 1

In this original research article by Graham et al, the development of LNCaP-derived CRPC cell lines that are either sensitive (M1, M2 and P1) or resistant (M1R1/R, M2R1/R, P1R1/R) to high dose testosterone are reported. The authors highlight response metrics of the cell lines to R1881 in vitro (clonogenic assays) and high dose testosterone in vivo (tumor volume and survival). Additionally, immunohistochemistry of AR expression from CRPC and HTR xenografts suggests that level of AR expression underpins the response to therapy. While the findings and generation of new preclinical models are potentially interesting, limited molecular characterization of the reported CRPC and HTR cell lines significantly detracts from the translatability of the models to the patient population and the synthesis of author conclusions. Below are comments and suggestions:

-Figure 1 is challenging to interpret given the information provided in the figure legend and Materials and Methods. Please clarify the cell lines in the clonogenic assay that were “supplemented with fetal bovine serum (FBS) or charcoal-stripped serum (CSS)" (Page 2, Line 81).  Were the assay parameters between the cell lines consistent? Additionally, please indicate the duration of R1881 treatments.

We have modified our methods sections to be more clear (lines 84-86). The assay parameters were identical between cell lines; the duration of R1881 treatments was at least 14 days; some cell lines were treated slightly longer than others based on rate of growth, but all were normalized to their own control (i.e. the 0 R1881 concentration group).

Comment 2: Please provide statistical analyses between treatment groups in Figure 1 to support your claims for all lines tested. The authors state that “The three CRPC cell lines all exhibited a dose-dependent reduction in colony growth with R1881” (page 4, Line 148). However, the M1 cell line appears to be stimulated by R1881 at 0.03 nM as reported in the M1R1/R cell line. The M2 cell line is exquisitely sensitive to R1881 with no apparent response differences between all R1881 concentrations tested.

We have performed statistical analysis between treatment groups in Figure 1. When analyzing the normalized data, statistical significance was not reached, likely due to the small sample size and reduced variability with normalization. We have added this to the text (lines 137-139 and 159-162). We agree there was nonsignificant stimulation by R1881 at 0.03nM in the M1 cell line and have modified the text to reflect this.

Comment 3: Please clarify the number of animals in each experimental group in figure legends or graphs for Figures 2-5. The reader must infer the number of animals in each of the groups from total number of animals reported in the main text. This has been added to the results section (Lines 180-181).

Comment 4: Figure 6, please describe the analytical tools or approaches used to determine cell numbers and estimate percentage of AR+ cells from IHC images. Are stromal cells excluded from the analyses or are they contributing to the data interpretations?

A GU pathologist reviewed all the H&E-stained slides and immunohistochemical stains. Tumor areas were randomly selected, photographed, and printed using a color printer. Tumor cells with and without AR positivity (approximately 5,000 cells per case) were manually counted. Malignant cells with AR positivity were graded based on staining density, and the percentage of each grade was calculated individually. Stromal cells were excluded from the study. We have added additional language to the methods section to make this more clear (Lines 129-131).

Comment 5: Culturing LNCaP cells in hormone depleted media, such as the P1 line, has been reported to induce cellular plasticity and neuroendocrine features (eg. PMID: 21227062; 27403603). LNCaP cells cultured in enzalutamide, such as M1 and M2, have been reported to express AR splice variants (eg LNCaP95 cells). Additional molecular phenotypes of AR+ and AR- CRPC have also been reported in the patient population (i.e. amphicrine, double-negative, AR-low). Moreover, the downregulation of AR protein expression observed in cells from the three HTR cell lines potentially indicates further transdifferentiation to other molecular subtypes or expression of AR-splice variants not detected by the IHC antibody. Do western blots, qPCR or IHC of neuroendocrine markers, EMT markers or AR-V7 and target genes provide more insights into molecular characteristics of CRPC lines reported here (P1, M1 and M2) or the mechanisms of resistance to high dose testosterone in the HTR lines (P1R1/R, M1R1/R and M2R1/R)?

The reviewer makes a good point. We have planned further molecular characterization of each of these resistant cell lines to further characterize these cells and explore potential mechanisms of resistance. However, on histologic examination these cells do not have morphologic characteristics of neuroendocrine/small cell cancer.

Reviewer 2 Report

Comments and Suggestions for Authors

The article of Laura S. Graham et al. “Cell Models of Castration Resistant and High Dose Testosterone Resistant Prostate Cancer Recapitulate the Heterogeneity of Response Observed in Clinical Practice” studied the mechanisms of castration resistance using original prostate cancer cell model and pharmacologic testing for clinical entity. Indeed, the treatment of metastatic castration resistant prostate cancer (CRPC), which is the main cause of high mortality from prostate cancer, still remains a complicated problem to solve emergency in current oncology. The main goal of this article was to understand the possible reasons of therapeutic resistance in both high dose testosterones naive CRPC and high dose testosterone resistant CRPC. To solve this problem, the authors have developed three cell models of high dose testosterone sensitive CRPC and high dose testosterone resistant CRPC which can relate to the heterogeneity seen in patients with CRPC. The authors confirmed a heterogeneous response to testosterone treatment in vitro and in vivo. They discussed that signaling by androgen receptors is assumed to play an important oncogenic role in the development of resistance to castration and, as a result, in the response to therapy, citing the article Ref. 8 Markowski MC and Ref.13. Isaacs J. et al., which were published in 2004 and 2012. Now days, many mechanisms of resistance are known including AR-dependent and -independent resistance mechanisms. See, for an example, Gaetano Aurili et al. Androgen Receptor Signaling Pathway in Prostate Cancer: From Genetics to Clinical Applications. Cells. 202010;9(12):2653. This needs to be discussed. Thus, the authors confirmed the heterogeneity of the response of the studied models to testosterone. In “Abstract” the authors declared that ..”Overall, we show that we have developed three models of HTR resistance that can be used to study mechanisms of high dose testosterone resistance and identify potential therapeutic targets”. However, in the “Discussion”, I did not see how the obtained results could help in the clinic. The question arises: can we take advantage of IHC analysis of AR as a prognosis factor? In this case, it is rather interesting to discuss how it can be determined in the absence of a primary tumor? I believe that after answering the above questions, this article can be published in the Cancers”.

Author Response

We thank the reviewers for taking the time to review our manuscript. We have responded to each review below in addition to including a manuscript with tracked changes.

Comment 1: The article of Laura S. Graham et al. “Cell Models of Castration Resistant and High Dose Testosterone Resistant Prostate Cancer Recapitulate the Heterogeneity of Response Observed in Clinical Practice” studied the mechanisms of castration resistance using original prostate cancer cell model and pharmacologic testing for clinical entity. Indeed, the treatment of metastatic castration resistant prostate cancer (CRPC), which is the main cause of high mortality from prostate cancer, still remains a complicated problem to solve emergency in current oncology. The main goal of this article was to understand the possible reasons of therapeutic resistance in both high dose testosterones naive CRPC and high dose testosterone resistant CRPC. To solve this problem, the authors have developed three cell models of high dose testosterone sensitive CRPC and high dose testosterone resistant CRPC which can relate to the heterogeneity seen in patients with CRPC. The authors confirmed a heterogeneous response to testosterone treatment in vitro and in vivo. They discussed that signaling by androgen receptors is assumed to play an important oncogenic role in the development of resistance to castration and, as a result, in the response to therapy, citing the article Ref. 8 Markowski MC and Ref.13. Isaacs J. et al., which were published in 2004 and 2012. Now days, many mechanisms of resistance are known including AR-dependent and -independent resistance mechanisms. See, for an example, Gaetano Auriliauril et al. Androgen Receptor Signaling Pathway in Prostate Cancer: From Genetics to Clinical Applications. Cells. 202010;9(12):2653. This needs to be discussed. Thus, the authors confirmed the heterogeneity of the response of the studied models to testosterone. In “Abstract” the authors declared that ..”Overall, we show that we have developed three models of HTR resistance that can be used to study mechanisms of high dose testosterone resistance and identify potential therapeutic targets”. However, in the “Discussion”, I did not see how the obtained results could help in the clinic. The question arises: can we take advantage of IHC analysis of AR as a prognosis factor? In this case, it is rather interesting to discuss how it can be determined in the absence of a primary tumor? I believe that after answering the above questions, this article can be published in the “Cancers”.

The reviewer has made many interesting points. We agree that there are many mechanisms of resistance known that lead to castration resistance, however, there is less known about what could lead to resistance to high dose testosterone. In this paper we believe we have developed high dose testosterone resistant cell lines that can help answer this question. We believe this will help in the clinic by identifying potential combination therapies for men pursuing high dose testosterone treatment. We have made this more clear in the discussion (Lines 320-326).